# Co-Infection of Tobacco Rattle and Cycas Necrotic Stunt Viruses in *Paeonia lactiflora*: Detection Strategies, Potential Origins of Infection, and Implications for *Paeonia* Germplasm Conservation

**DOI:** 10.3390/v16060893

**Published:** 2024-05-31

**Authors:** Nastassia B. Vlasava, David C. Michener, Siarhei Kharytonchyk, Liliana Cortés-Ortiz

**Affiliations:** 1Matthaei Botanical Gardens and Nichols Arboretum, University of Michigan, Ann Arbor, MI 48105, USA; michener@umich.edu; 2Central Botanical Garden, National Academy of Sciences of Belarus, 220012 Minsk, Belarus; 3Department of Microbiology and Immunology, University of Michigan Medical School, Ann Arbor, MI 48109, USA; siarheik@umich.edu; 4Department of Ecology and Evolutionary Biology, University of Michigan, Ann Arbor, MI 48109, USA; lcortes@umich.edu

**Keywords:** peonies, genetic diversity, phylogenetics, mix viral infection, genetic resources, germplasm management

## Abstract

Increasing reports of tobacco rattle virus (TRV) and cycas necrotic stunt virus (CNSV) in herbaceous *Paeonia* worldwide highlight the importance of conserving the genetic resources of this economically important ornamental and medicinal crop. The unknown origin(s) of infection, differential susceptibility of peony cultivars to these viruses, and elusive disease phenotypes for CNSV in peonies make early detection and management challenging. Here, we report the presence of TRV and CNSV in plants of the University of Michigan living peony collection in the United States and a molecular characterization of their strains. Using sequences of the TRV 194 K RNA polymerase gene, we confirmed TRV infections in seven symptomatic plants (1.07% of all plants in the collection). Using newly developed primers, we recovered sequences of the CNSV RdRp gene and the polyprotein 1 gene region from nine out of twelve samples analyzed, including three from symptomless plants. Four of the nine plants had TRV and CNSV co-infections and showed more severe disease symptoms than plants only infected with TRV. Phylogenetic analyses of isolates from the University of Michigan living peony collection and publicly available isolates point to multiple origins of TRV and CNSV infections in this collection. This is the first report of TRV/CNSV co-infection and of a symptomatic detection of CNSV on cultivated *P. lactiflora*.

## 1. Introduction

Viruses represent serious threats to plant germplasm and living collections, because they can quickly spread if not detected and managed properly. Collections of living plants that are vegetatively propagated are particularly susceptible to this risk [1,2] because viral infections can persist or accumulate in plants over each generation, and can be further dispersed to healthy plants via stock distribution, viral vectors, and/or mechanical transmission. The risk is not trivial: ornamental plants often host viruses that can infect other horticultural and agricultural crops, as well as species of native flora [3]. Herbaceous peonies are perennial crops of significant ornamental and medicinal value [4,5,6] that are mainly propagated vegetatively, and are thus susceptible to the scenario above. However, there is limited information on peony viral pathogens’ control, their transmission, and their distribution at local and global scales [7].

Herbaceous peonies (primarily *Paeonia lactiflora* Pall. cultivars) have been domesticated for over four millennia [4,5,6]. This long history of domestication of *P. lactiflora*, including crosses with other *Paeonia* species, has led to over 4500 documented cultivars [6,8], most of which are no longer commercially available. From the total number of peony cultivars registered by the American Peony Society, more than half are either extinct or only remain in isolated living collections and public gardens [9]. The University of Michigan’s Nichols Arboretum (UMNA) holds one of the largest reference collections of historic herbaceous peonies in United States and is an invaluable reservoir of genetic diversity of cultivated *Paeonia*, as nearly half the accessions are not found in commerce or in any other collections. Keeping these collections free of pathogenic viruses is crucial to reduce the risk of extinction of those historic cultivars and their unique genetic diversity. Identifying viral agents as part of the health management practices of ornamental plant collections is fundamental to effectively detect and control potential outbreaks of viral diseases [3,10,11,12,13].

Over the last decade, there has been an increasing number of reports of viral diseases in cultivated peonies, including new viral pathogens and emergent infections in a broad geographical range [14,15,16,17,18,19,20,21,22,23,24]. Among them, tobacco rattle virus (TRV, *Tobravirus*, Virgaviridae) is often reported in cultivated peonies worldwide, since its first report in peonies in New Zealand (reviewed in [25]) and Japan [26] in the middle of 20th century. TRV causes disease on a broad range of plant species, producing severe economic losses in potatoes [27] and other crops (e.g., flower bulbs [28]). TRV has a linear single-stranded bipartite positive-sense RNA genome [29,30], is transmitted by nematodes (*Trichodorus* and *Paratrichodorus* spp.) as the primary vector [31], and causes reduced fitness in the host. For example, TRV infection in potatoes causes tuber necrosis, which may significantly reduce the yield quality [27,32]. In the United States, TRV has been reported in peonies in Alaska [19], Ohio [14], and Washington (unpublished, GenBank accession number JX315458). Owen et al. [33] reported a TRV infection in Michigan peonies using a molecular test, but sequences of the viral isolates are not publicly available. Therefore, it remains unknown what TRV viral isolates have infected Michigan peonies and their relationship to those reported in other locations. Given the molecular diversity of viral isolates observed in other crops within and between locations, identifying the TRV isolates infecting peonies in Michigan is critical to infer the potential sources of infection and control their spread.

Cycas necrotic stunt virus (CNSV, *Nepovirus*, Secoviridae) is another virus that infects peonies. CNSV has a linear single-stranded bipartite positive-sense RNA genome and belongs to the Nepovirus subgroup B [34,35,36]. In *Paeonia*, it has been isolated from *P. lactiflora* [20,23,37], *P. suffruticosa* [17,18], and *P. officinalis* × *P. lactiflora* [24]. Since its discovery in a cycad species (*Cycas revoluta*) in Japan [36], CNSV has been found in angiosperms over a broad geographic range, including New Zealand, Australia, South Korea, Japan, and China [17,18,23,24,37,38,39,40,41]. Besides peonies, CNSV has been isolated from other angiosperm ornamental crops such as daphne (*Daphne odora*, Thymeliaceae, [18,40]), as well as monocotyledonous gladiolus (*Gladiolus* spp., Iridaceae, [38]), and Easter lilies (*Lilium longiflorum*, Liliaceae, [41]). Some reports mention that plants with confirmed CNSV infections usually show stuntedness [17,20,23]. However, until this report, no clear description of disease symptoms has been associated with CNSV infection in peonies [34,40]. We recently reported the presence of CNSV in the continental United States, which was isolated from peonies grown in the UMNA collection (Michigan), as well as in Alaska, Arkansas, and Oregon [42], which represent the first report of CNSV in the western hemisphere [20]. Similar to TRV, evaluating the extent of variation in CNSV isolates in Michigan peonies and their relationship to isolates infecting other cultivars in different geographic regions can provide insight into the number and sources of infection in this collection and broaden the geographic representation for the phylogenetic analysis of this virus. Understanding the epidemiology of mixed infections is important because the simultaneous infections of multiple viruses in plants can exacerbate disease severity for the host, when compared to single-virus infections [43,44,45,46]. Recently, CNSV co-infections with multiple viruses have been reported in tree peonies [17]. We have previously suggested that the Lemoine Disease of herbaceous peonies, which has an unresolved etiology but produces general stuntedness in the plant and a distortion of root growth [15], might be caused by a mixed infection [20], but its connection with the presence of CNSV remains unexplored.

The goals of this study were to investigate the infection of TRV and CNSV (and their co-infection) in the multi-cultivar *Paeonia* collection of the UMNA (hereafter, UMNA *Paeonia* reference collection), analyze the pathogen diversity associated with different peony hosts (cultivars), and use a phylogenetic framework to infer the origins of these viral infections. We combined our molecular analysis of viral infection in plant specimens with a multiyear (2015–2020) observational study of plant symptoms in the UMNA *Paeonia* reference collection to address the following specific objectives: (1) optimize the detection strategy and detect the presence of TRV and CNSV in different cultivars of *P. lactiflora*; (2) characterize the molecular diversity of TRV and CNSV isolated in Michigan and assess their relationships with reported isolates of these viruses elsewhere to infer their potential origins; and (3) investigate the co-infection of TRV and CNSV in peonies and its implications for germplasm collections. We conclude that the TRV and CNSV infections in the UMNA *Paeonia* reference collection likely have multiple origins, CNSV infections may go unnoticed in asymptomatic specimens, and the co-infection of TRV and CNSV in the same plant host produced more severe symptoms than single infections of each virus. Future work to understand why only a few cultivars are infected despite them sharing close proximity with many other cultivars may provide an interesting platform on which to investigate viral transmission and resistance mechanisms to viral infections in peonies.

## 2. Materials and Methods

### 2.1. The UMNA Peony Reference Collection

The UMNA living *Paeonia* collection was established in 1922 and is an active reference field collection of peony cultivars and a public display garden (Ann Arbor, MI, USA, 42.28° N, −83.73° W). Between 2015 and 2020, we monitored 584 out of 799 living specimens of herbaceous peonies belonging to 372 named *P. lactiflora* cultivars, 37 unnamed cultivars, and hybrids of unknown origin. The peony plants are maintained in beds, and each cultivar is represented by two accessions (i.e., biological replicates). The specific location of each accession has been repeatedly mapped since 1927, and there are individuals that have never been moved since they were originally planted. No chemical treatments (i.e., for weed or disease control, or fertilizers) are recorded to have been used since the 1930s until today.

### 2.2. Sampling, RNA Extraction, Reverse Transcription, and PCR Amplification of Viral Genes

In 2015 and 2016, we sampled 1–3 leaves of each symptomatic TRV plant (N = 6, 1 of which was sampled in both years) and 5 asymptomatic plants that served as controls. Leaves were placed on dry ice immediately after collection and were kept frozen during transportation to the lab, where they were stored at −80 °C until they were processed. We extracted total RNA of all collected leaves per plant with the RNeasy Plant Mini Kit (Qiagen, Valencia, CA, USA) according to manufacturer’s protocols. Briefly, we disrupted ~100 mg of freshly frozen tissue with a mortar and pestle in liquid nitrogen, grinding it to a fine powder before proceeding with the extraction protocol. We then mixed the powdered sample with 450 μL Buffer RLT, containing 0.01% of β-mercaptoethanol, and incubated it at 56 °C for 3 min to lysate the sample. We continued with all the manufacturer’s protocol steps using the provided QIAshredder spin column, the RNeasy spin column, and the kit reagents. We eluted the DNA in 30 μL of ultrapure water and stored them at −20 °C until use. We then synthesized first-strand cDNA using the RevertAid First Strand cDNA Synthesis Kit (Thermo Scientific Inc., Waltham, MA, USA) with random hexamer primers, using the total RNA as a template. Prior to the reverse transcription reaction, RNA was incubated at 82 °C for 5 min, followed by chilling on ice. We then mixed 3 μL (~20 ng) of purified RNA with 1 μL of random hexamer primer, 4 μL 5× Reaction Buffer, 1 μL of RiboLock RNase Inhibitor (20 U/μL), 2 μL of 10 mM dNTP mix, 1 μL of 200 U/μL RevertAid M-MuLV RT (Thermo Scientific™), and 8 μL of ultrapure water for a final reaction volume of 20 μL. The reaction was incubated for 5 min at 25 °C, followed by 60 min at 42 °C and terminated by heating at 70 °C for 5 min. We included both a negative (reaction without an RNA template) control and a positive (human GAPDH RNA and GAPDH-specific PCR primers) control, provided in the kit. For the molecular identification of TRV, we used specific TRV primers [14] (Table 1), which are expected to amplify a fragment of 779 bp size of the TRV 194 K RNA polymerase gene. All reactions included negative controls. Independently, the identification of TRV was also conducted by Agdia Inc. (Elkhart, IN, USA) using real-time PCR under the company’s protocols to detect and quantify the TRV 16-Kilodalton gene (TRV 16-K) that encodes the 16 kDa putative RNA silencing suppressor.

For several samples, our TRV amplification showed a band of a size smaller than expected: ~450 bp. We isolated this fragment by excising the band and cloning it into the pGEM-T Easy Vector (Promega, Madison, WI, USA) using the manufacturer’s protocols. We then sequenced clones with Sanger sequencing at the University of Michigan sequencing core. The obtained sequences (GenBank accessions MK521453 and MK521454) showed an 85.5% nucleotide similarity with the polyprotein 1 gene (positions 4328–4772) of a CNSV Lily1 isolate from Japan (GenBank accession number JN127336) and a 91.5–95.0% amino acid similarity with the RdRp (RNA-dependent RNA polymerase) region of polyprotein 1 gene from CNSV isolates from Japan and Australia (GenBank accession numbers AEN25475, NP_620619, and NP_734016). We used the newly generated sequences to design new specific primers (Table 1) to amplify a 382 bp segment of CNSV polyprotein 1 gene (the 4349–4730 bp region of the gene based on homology to the GenBank accession number JN127336). We conducted PCR amplifications for all samples (N = 12) collected between 2015 and 2016 using this new primer pair. To independently confirm the presence of CNSV in these plants, we also performed a reverse transcription PCR with two sets of primers specific to the RdRp region (Table 1) that discriminate between Nepovirus subgroups A and B, respectively [47]. All PCR reactions were conducted in a 25 μL volume using Platinum Taq DNA polymerase (5U/μL, Invitrogen, Carlsbad, CA, USA) and included negative controls (see Table 1).

### 2.3. DNA Sequencing

We sequenced PCR products using Sanger sequencing (University of Michigan Sequencing Core, Ann Arbor, MI, USA). We inspected chromatograms for accuracy using 4Peaks (Nucleobytes, The Netherlands) and DNASTAR (Lasergene Genomics, Maddison, WI, USA). We confirmed the identity of each sequence using BLAST or Blastx in GenBank (http://www.ncbi.nlm.nih.gov, accessed on 15 October 2020). We deposited all new sequences in GenBank (accession numbers MF918561-918567 for TRV and MK521443-521454 and MK493788-493796 for CNSV).

### 2.4. Diversity and Phylogenetic Relationships among Virus Isolates

We used a Maximum Likelihood phylogenetic analysis to determine the diversity of TRV and CNSV isolates in the UMNA *Paeonia* reference collection, and to assess their relationship to published TRV and CNSV isolates. For the CNSV phylogeny, we concatenated sequences of the RdRp gene and the amplified region of the CNSV polyprotein 1 gene. We used sequences of the Pea Early Browning Virus (GenBank accession X14006) and the Artichoke Italian Latent Virus isolate AILV-V (GenBank accession NC_043684) as outgroups for TRV and CNSV phylogenetic reconstructions, respectively. We used ClustalW [48] to independently align our TRV and CNSV sequences with all publicly available sequences in GenBank for the respective genes (Appendix A) and manually edited them to ensure proper alignments. Using MEGA v. 10 [49], we determined the best evolutionary models for each phylogenetic reconstruction, built the phylogenies, and assessed the statistical support for each topology through 1000 bootstrap permutations [50].

### 2.5. Visual Detection of TRV and CNSV Infection

From 2015 to 2024, we recorded the presence/absence of visual symptoms of TRV infection, such as leaves with yellow or light-green mottle mosaic, concentric ringspots, and line patterns (Figure 1A–C) for each plant of the UMNA *Paeonia* collection. These symptoms were present on the surface of the leaves of infected plants regardless of their shape. Leaves with symptoms were mature but not old, and usually located in the middle level of the plant (between the ground and the top of the plant, Figure 1C). We estimated the severity of the TRV disease based on a score of leaves with symptoms, using scores from 0 to 5, with 0 representing plants with no symptoms in any leaves and 5 representing plants with multiple leaves displaying symptoms of severe mottle mosaic according to [51]. This assessment was conducted after plants began blooming and the leaves were mature but not senesced, which is usually consistent with the peak of symptoms. To assess symptoms of disease due to CNSV infection, we reviewed photographs of all plants positive for CNSV in 2015 and 2016 and the observational data generated to evaluate TRV infections.

## 3. Results

### 3.1. Molecular Confirmation of TRV in Symptomatic Peony Plants

Only 6 out of 584 screened plants at the UMNA collection in 2015 and 2016 showed symptoms of TRV infection, which were consistently observed in the same symptomatic plants over the following years (up to 2024, Appendix A). Samples of all symptomatic plants amplified the TRV 194 K RNA polymerase gene (Table 2, Appendix A), and the sequences (GenBank accessions MF918561–MF918567) shared 96–100% nucleotide sequence similarity with each other, and 91–100% with publicly available TRV sequences. The Agdia Inc. real-time PCR independently confirmed the presence of TRV for the same plants that were positive in our PCR amplification (Table 2).

### 3.2. TRV Diversity at the UMNA Peony Collection

The TRV phylogenetic tree of the UMNA peony sequences and other publicly available TRV sequences (Appendix A) shows two major clades (labeled as TRV I and TRV II in Figure 2). Clade TRV I includes mainly isolates from Europe, and a few isolates from Ohio (SOS6-TRV, SOS8-TRV, and SOS8-Tobra, isolated from *Hosta* sp. cultivars) and Alaska (1AKBH from *Dicentra spectabilis*) (Appendix A), but no sequences isolated from the UMNA peonies. Clade TRV II comprises isolates from Europe, Asia, and the United States, and includes all sequences from isolates of UMNA peonies (Figure 2). Sequences of isolates generated in this study from four different cultivars show between 96 and 99% nucleotide similarity among them, and cluster in three different subclades. One of our sequences grouped closely with sequences isolated from *P. lactiflora* and *P. suffruticosa* from China, *Nicotiana clevelandii* from Oregon, isolates from several other crops from Europe, and two accessions isolated from potatoes in Africa (Figure 2, Appendix A). In another subclade, our TRV sequences were closely related to those isolated from *P. lactiflora* in Ohio, a sequence isolated from Michigan potatoes (MI-1), several European isolates, and one isolate from *P. lactiflora* from Japan (Figure 2, Appendix A). All our remaining TRV isolates grouped together in the third subclade, together with 23 sequences isolated from *Dicentra spectabilis* (bleeding heart) from Ohio [52].

### 3.3. Molecular Identification of CNSV and Co-Infection with TRV in the UMNA Peony Collection

Using our newly designed primers, we detected CNSV infection in 9 out of 12 plant samples (75%; Table 2) collected in 2015 and 2016, including some symptomless plants that we had originally used as negative controls for TRV identification (Figure 1D and Appendix A). CNSV sequences (GenBank accession numbers MK493788–MK493796) displayed 80–90% nucleotide sequence similarity and 94–97% amino acid sequence similarity with other CNSV isolates available in GenBank (accession numbers AB073147, JN127336, EU741695, MK512741, and MK521837, for nucleotide sequences, and AEN25475 for amino acid sequences). We did not obtain any PCR products with the Nepovirus A-specific primers, but the Nepovirus B-specific primers (Table 1) amplified a fragment of the expected size (~250 bp) in the same plants that were positive with our specific Nepo_Pl primers (Table 2, Appendix A). The Nepovirus B primers amplify a different region of CNSV RNA1—RdRp than the Nepo_Pl primers we designed, thus independently confirming the presence of the CNSV virus in our samples. All sequences were deposited in GenBank (accession numbers MK521443–MK521452). The nucleotide sequence similarity of the RdRp fragment to other CNSV sequences from GenBank (Appendix A) was between 81 and 100% (90–100% amino acid sequence similarity). Two of the five samples that were negative controls (i.e., did not show any visual symptoms of disease) for our TRV analysis were molecularly confirmed to be infected with CNSV. Furthermore, five plants had TRV and CNSV co-infections (71% of all TRV-infected plants, Table 2).

### 3.4. Michigan CNSV Isolates Display Close Relationship to Several Paeonia Isolates from Asia

All CNSV nucleotide sequences isolated from our nine samples were different from each other (Figure 3). For the region amplified with our newly designed Nepo_Pl primers, nucleotide substitutions among sequences did not affect amino acid products (i.e., were synonymous), but for the region amplified with the Nepovirus B primers nucleotide sequence variants resulted in two RdRp amino acid sequences that differed in two amino acids: Gly/Ser^1838^ and Ala/Thr^1867^ (Figure 4). Although we only detected one amino acid sequence in each plant, and the two biological replicates usually had the same sequence, the two plants of the Gisele cultivar had different amino acid sequences (Table 2, Figure 4).

The CNSV phylogenetic tree grouped sequences of all isolates from UMNA in two well-supported clades (labeled CNSV I and CNSV II in Figure 3). Clade CNSV I contained five isolates from Michigan and sequences isolated from *P. suffruticosa* (AD and Anhiu isolates) and *P. officinalis* (PO) from South Korea, and *P. lactiflora* (BJ) from China and other USA locations (Oregon, New York, and Alaska; Figure 3). Clade CNSV II contained sequences isolated from *P. lactiflora* from Japan, China, and the USA (Arkansas and New York). These two clades were separated from another containing one isolate from *P. lactiflora* from New Zealand and isolates from different plant species in Australia, China, Japan, and South Korea (Clade CNSV III, Figure 3).

### 3.5. Visual Indicators of CNSV Infection and TRV/CNSV Co-Infection on P. lactiflora Plants

We discovered the presence of CNSV using molecular approaches from peony plants that had no visual evidence of disease symptoms. We used our ongoing observations of phenotypic symptoms of the UMNA plants over five years (2015–2020) to assess the potential effect of the CNSV infection in these plants. Two plants for which infection with CNSV was molecularly confirmed were partially or uniformly stunted, although they did not show symptoms characteristic of TRV. Plants infected with CNSV also showed interveinal chlorotic mottle areas (faint to bright) scattered in their foliage, and two plants also presented purple-red spots and necrotic areas close to the edges of the leaves (Figure 5), and/or slightly puckered and uneven distortions of leaf surfaces. Plants confirmed to have both TRV and CNSV infections showed the CNSV symptoms described above, as well as more severe TRV symptoms (area of lesions on leaves and brightness of chlorotic mosaics) than those that only were positive for TRV (Table 2). In total, the molecular confirmation of TRV and CNSV infection and the long-term monitoring of symptoms of plants in the collection allowed us to verify that, out of 281 cultivars, 4 (Duchess of Portland, Gisele, Yeso, and Gigantea) are susceptible to TRV, at least 6 (Duchess of Portland, Gisele, Yeso, Gigantea, Jeannot, and Isoline) to CNSV, and 3 (Gisele, Yeso, and Gigantea) have co-infections of both viruses (Table 2).

## 4. Discussion

Pathogenic viruses threaten plant germplasm collections, particularly those of vegetatively propagated plants [1,19], such as peonies [14,16,18,20,21,22,54]. In this study, we provide molecular evidence of infection of TRV (which is widely documented elsewhere in peonies) and of the newly emerging CNSV for the United States [20,42,55] in the UMNA *Paeonia* reference collection in Michigan (USA). We also provide evidence of TRV/CNSV co-infection in some specimens. To our knowledge, co-infections of TRV and CNSV had not been previously reported for peonies or in any other taxa. We also used a phylogenetic approach to evaluate the diversity of isolates for the two viruses and to assess the relationship of the UMNA isolates with respect to those reported in other parts of the world to infer possible origins of these isolates.

### 4.1. Diversity of TRV and CNSV Isolates in Michigan

Our phylogenetic analyses revealed variation in both TRV and CNSV isolated from plants of different cultivars, the same cultivar, and even between samples of the same plant collected in different years (Figure 2 and Figure 3). Our TRV analysis included a larger number of sequences than any previous study, providing a higher resolution for the placement of our TRV isolates from Michigan peonies, and overall was consistent with previous phylogenetic analyses [27,56]. Sequences isolated from UMNA peonies belong to three TRV lineages. Some sequences clustered with those isolated from other *Paeonia* species and cultivars (e.g., those in China and in the US), but other sequences clustered with isolates reported from other species that are not taxonomically related, but geographically proximate. For example, TRV sequences (MF918562 and MF918563) from Michigan peonies are equally related to those isolated from Michigan potatoes, Ohio peonies, *Lysimachia nummularia* from Finland, and *P. lactiflora* from Japan (Figure 2), showing that these lineages exist in very distant locations and are not host-specific.

The distant transmission of TRV to UMNA peonies is consistent with the routine exchange/trade of vegetative propagules of peonies across the world, in which viruses can persist. There is also the possibility of short-distance TRV transmission from *Paeonia* and other genera (i.e., *Dicentra*, *Hosta*, potato, etc.), based on the close phylogenetic relationship of these isolates to those from UMNA peonies. Overall, our results provide evidence that there are at least three distinct lineages of TRV in the peony collection at UMNA, and suggest multiple origins of TRV in Michigan, which could have occurred through the introduction of infected *Paeonia* specimens or from the transmission (possibly by nematodes [57,58]) from other plant species, as this collection resides within a university garden context.

Phylogenetic analyses confirmed that CNSV sequences isolated from UMNA peonies are most closely related to those isolated from *Paeonia* cultivars from China, Japan, South Korea, and other locations in the USA (but not from those isolated in *P. lactiflora* from New Zealand, which is one of the earliest characterized isolates). Despite the recent discovery of CNSV in the US [20,42,55], it is unclear when this virus arrived, given that it could have been undetected due to the lack of clear disease symptoms, and because the use of high-resolution molecular methods to detect its presence is relatively recent (e.g., [17,18,20,23,37,42] and this study). Despite the limited availability of previously published sequences for the CNSV genomic regions that we amplified in this study, our phylogenetic analysis suggests the CNSV isolates from UMNA peonies might be host-specific for *Paeonia*, given that closely related isolates have only been found in peonies in Asia and the USA. The phylogenetic proximity of USA and Asian CNSV isolates might be explained by the active germplasm exchange of peony vegetative propagules domestically and with important historic breeding centers in Asia.

### 4.2. Symptoms of Peonies with CNSV and TRV Infection and Co-Infection and Implications for the Collection Management

Our molecular confirmation of TRV infection in all plants that showed visual symptoms of TRV disease demonstrates the power of the visual diagnostics of this virus and reveals that the current incidence of TRV in the UMNA is low (1.07%). TRV affects the physiological well-being of the infected *Paeonia* plants, threatening their long-term fitness [15] and their ornamental value. They may also become a source of further infection to other plants in the collection or of native species in the area. However, there is a lack of data on the resistance of the specific cultivars to viruses. TRV isolates’ virulence or host plant genes of resistance are not characterized. Future genomic studies on infected and non-infected plants in the collection may provide an opportunity to understand the genetic mechanisms underlying resistance.

Our results also revealed a high incidence of CNSV infection (75%) in the subset of plants analyzed, although some lacked visual symptoms of disease. While some plants molecularly confirmed to host CNSV were completely asymptomatic, cultivars with TRV and CNSV co-infection showed amplified visual symptoms of disease. The symptoms we observed in the UMNA peonies infected with CNSV are similar to those reported for *P. lactiflora* with CNSV infection in New Zealand [23], and for tree peonies co-infected with CNSV and grapevine line pattern virus (GLPV) [17]. However, the foliar symptoms observed across all CNSV-infected cultivars were not equally developed, not even among biological replicates (plants of the same cultivar) planted adjacent to each other, which may reflect heterogeneity in viral properties, as reported among four CNSV isolates from gladiolus [38]. The roots of CNSV-infected plants at UMNA revealed symptoms like those described for Lemoine Disease in peonies [15], specifically, irregular swellings on tuberous and fine roots. The etiology of the Lemoine Disease is unknown [15,55], and thus its association with CNSV infection remains to be investigated.

Although mixed viral infections of CNSV with other viruses have been previously reported [17,22,41,59], to our knowledge, no previous reports have found CNSV and TRV simultaneously infecting the same plant, and the physiological impact of such co-infection is unknown. Therefore, further investigations on the impact of co-infection of CNSV and TRV in peonies is needed, especially considering reports that the co-infection of multiple viruses in plants can lead to high disease prevalence and an increase in the probability of virus recombination and evolution [44,45,60]. We urge the development of integrative management strategies and protocols to help maintain virus-free collections of *Paeonia* germplasm across institutions (we provide a brief description of such strategies in Appendix A).

## 5. Conclusions

In this study, we report the first molecular characterization of TRV isolated from peonies in Michigan and the first report of the co-infection of TRV and CNSV. This first report of the co-infection of TRV and CNSV in peonies shows how the simultaneous infection of multiple viruses may exacerbate disease symptoms in this economically important ornamental plant, and highlights the importance of the continuous monitoring of the living collection at the UMNA to avoid the spread of detrimental viruses. It also opens the possibility that some observed diseases with unknown etiology, such as the Lemoine Disease, may be caused by multiple viruses simultaneously affecting a host. Further experimental studies are needed to understand the extent to which the co-infection of TRV and CNSV impact current commercial peony cultivars.

We found that both TRV and CNSV detected in the UMNA peony collection likely have multiple origins. TRV has several lineages that cluster with lineages isolated in Europe, Asia, and USA, which are likely derived from other species of plants. Michigan CNSV isolates are most closely related to sequences isolated from *Paeonia* in Japan, China, and South Korea, providing evidence of the genus-specific transmission of the virus, and possibly from Asian countries, where, historically, the breeding and propagation centers of this crop are located [5]. We described potential symptoms of CNSV infection in *P. lactiflora* plants and reported on the more severe manifestation of disease in plants that were co-infected with TRV and CNSV, compared to single TRV infections. The results of our work highlight the threat that TRV and CNSV infections may pose, not only to peony germplasm collections, but also to the economically important industry of cultivated peonies, and call attention to the importance of carefully monitoring the exchange/trade of plant materials.

## Figures and Tables

**Figure 1 viruses-16-00893-f001:**
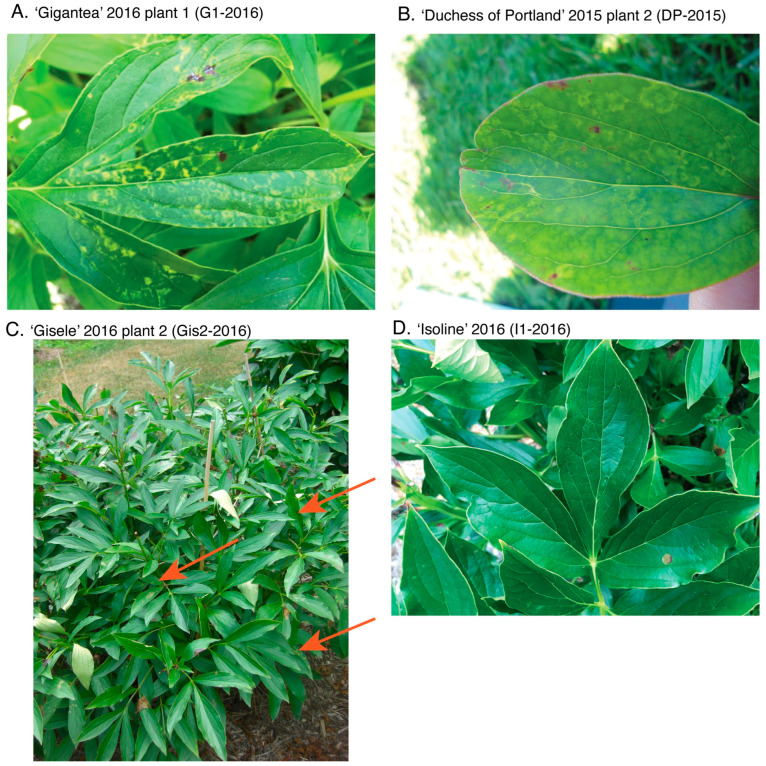
Examples of *Paeonia lactiflora* cultivars from the UMNA peony living collection exhibiting TRV symptoms. (**A**,**B**) Leaf mottle mosaic and concentric yellow ringspots in two different cultivars. (**C**) Symptomatic peony plant showing the area of the plant where leaves commonly show TRV symptoms. Red arrows point at some leaves with TRV symptoms. (**D**) Symptomless plant negative for TRV (but positive for CNSV). The year after the name of the cultivar refers to when the samples were collected, and when photographs were taken. In parenthesis is the abbreviation of the plant identification, as it appears on Table 2.

**Figure 2 viruses-16-00893-f002:**
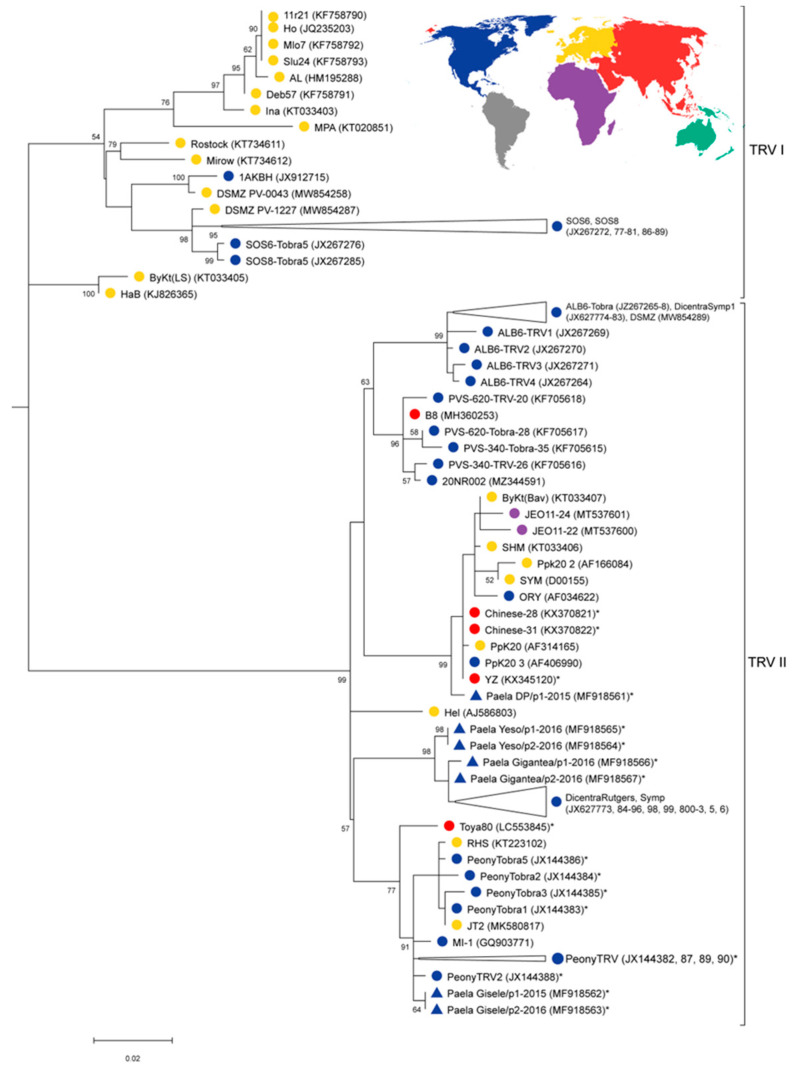
Maximum Likelihood phylogenetic tree depicting the relationships of TRV sequences isolated from peonies at the UMNA (blue triangles) and TRV sequences publicly available on GenBank, based on 799 bp of the RNA-polymerase gene region. The tree was built using the Tamura–Nei model [53] with invariable sites and gamma distribution (G + I). The Pea Early Browning Virus (PEBV) was used as the outgroup, and all nucleotide sites were included in the analysis. The numbers next to the nodes represent the branch support (1000 bootstrap replicates). Sequences isolated from the *Paeonia* genus (*P. lactiflora* or *P. suffruticosa*) are indicated with an asterisk (*). The color labels correspond to the broad geographical location of samples on the map. Sequences in some closely related clusters were collapsed, but all accession numbers are included in the label.

**Figure 3 viruses-16-00893-f003:**
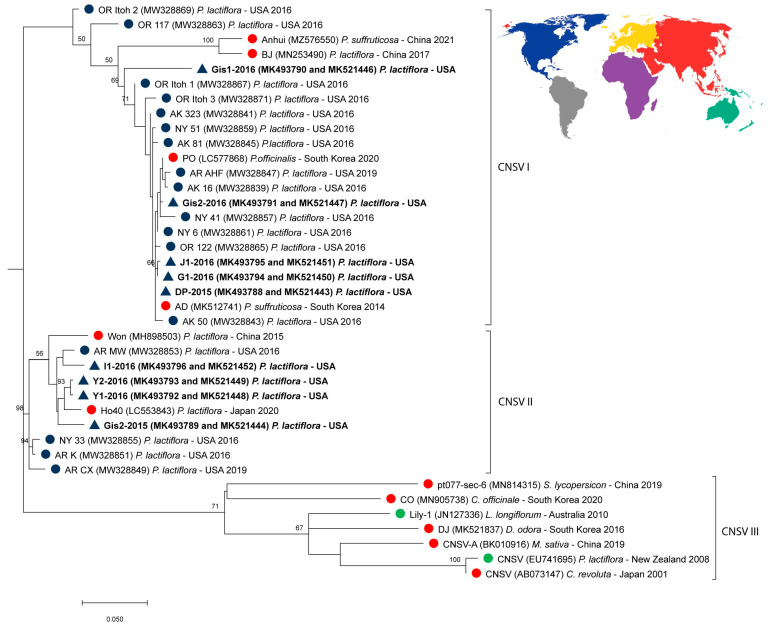
Maximum Likelihood phylogenetic tree depicting the relationships of *Cycas necrotic* stunt virus (CNSV) sequences isolated from peonies in the UMNA (blue triangles and in bold) and those available in GenBank (Appendix A) based on concatenated nucleotide sequences (534 bp) of fragments of the RdRp and polyprotein 1 genes. The tree was inferred using the Tamura–Nei model of sequence evolution [53] with invariable sites and gamma distribution (G + I). All nucleotide sites were used in the analysis. The numbers on nodes represent the statistical branch support (1000 bootstrap replicates). The Artichoke Italian latent virus (AILV) was used as an outgroup.

**Figure 4 viruses-16-00893-f004:**
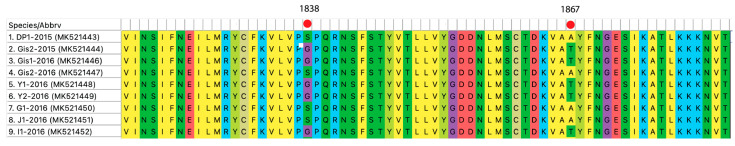
Alignment of the amino acid sequences of the CNSV RdRp region isolated from Michigan peonies. Positions of non-synonymous changes between the sequences are marked with red dots. Names correspond to abbreviations shown in Table 2, with GenBank accession numbers in between brackets.

**Figure 5 viruses-16-00893-f005:**
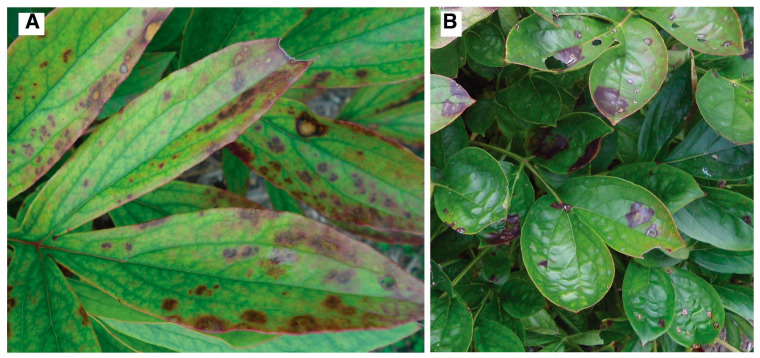
Interveinal chlorotic mottle, purple-red spots (**A**), and distortion of leaflet surfaces (**B**) observed on the leaves of CNSV-infected *P. lactiflora* cultivars (from the UMNA peony living collection).

**Table 1 viruses-16-00893-t001:** Primers used for the reverse transcription PCR detection of TRV and CNSV in *P. lactiflora* plants.

Virus (Abbreviation)	Primer Name, Sequence (5′-3′), and Reference	Genomic Region	Annealing T (°C) in This Study	Amplicon Size (bp)	PCR Conditions
Tobacco Rattle Virus (TRV)	TRVF683 (forward) 5′GCTATTGGTGATCAAGCTAGAAG3′ and TRVR1439 5′GCHGCCCCGTTWATGAAYARGAC3′ (reverse) [14]	194 K RNA polymerase gene	52	779	In total, 25 μL reaction (0.5 units of Platinum Taq DNA polymerase (Invitrogen, Carlsbad, CA, USA), 2.5 mM MgCl_2_, 0.2 mM dNTP mix, 0.2 μM of each primer, and 1 μL of cDNA). The cycling conditions: 94 °C (2 min), 30 cycles of 94 °C (45 s), 52 °C (30 s), 72 °C (60 s), and a final extension 72 °C (10 min)
Cycas Necrotic Stunt Virus (CNSV)	TRVF683 (forward) 5′GCTATTGGTGATCAAGCTAGAAG-3′ and TRVR1439 5′GCHGCCCCGTTWATGAAYARGAC3′ (reverse) [14]	polyprotein 1 gene, partial	52	~450	As above
Cycas Necrotic Stunt Virus (CNSV)	NepoPl_F 5′CTATTCTTCTTGGGCAAATGGGGTG3′ andNepoPl_R 5′GACCTGACTCCACTGCATTTCATTATG3′ [This study]	4349–4730 nt region of CNSV polyprotein 1	52	380	In total, 25 μL reaction (0.5 units of Platinum Taq DNA polymerase (Invitrogen, Carlsbad, CA, USA), 1.5 mM MgCl_2_, 0.2 mM dNTP mix, 0.2 μM of each primer, and 2 μL of cDNA). The cycling conditions: 94 °C (2 min), 30 cycles of 94 °C (30 s), 52 °C (30 s), 72 °C (60 s), and a final extension 72 °C (10 min)
Cycas Necrotic Stunt Virus (CNSV)	NepoB-F (forward) 5′TCTGGITTTGCYTTRACRGT3′ NepoB-R (reverse) 5′CTTRTCACTVCCATCRGTAA3′ [47]	RdRp gene	50	250	In total, 25 μL reaction (0.5 units of Platinum Taq DNA polymerase (Invitrogen, Carlsbad, CA, USA), 1.5 mM MgCl_2_, 0.2 mM dNTP mix, 0.2 μM of each primer, and 2 μL of cDNA). The cycling conditions: 94 °C (5 min), 35 cycles of 94 °C (30 s), 50 °C (30 s), 65 °C (2 min), and a final extension 72 °C (5 min)
*Cycas Necrotic Stunt Virus* (CNSV)	NepoA-F 5′ACDTCWGARGGITAYCC3′ (forward) NepoA-R 5′RATDCCYACYTGRCWIGGCA3′ (reverse) [47]	RdRp gene	50	340	In total, 25 μL reaction (0.5 units of Platinum Taq DNA polymerase (Invitrogen, Carlsbad, CA, USA), 1.5 mM MgCl_2_, 0.2 mM dNTP mix, 0.2 μM of each primer, 2 μL of cDNA). The cycling conditions: 94 °C (2 min), 35 cycles of 94 °C (30 s), 50 °C (30 s), 69 °C (2 min), and a final extension 72 °C (5 min)

**Table 2 viruses-16-00893-t002:** Results of presence/absence of amplicons for each reverse transcription PCR of TRV and CNSV in *P. lactiflora* plants using different primer combinations, and scores of TRV symptoms for each plant. Plus (+) and minus (−) signs represent positive and negative amplifications, respectively. GenBank accession numbers of sequences of positive amplifications are: TRV 94 K RNA polymerase gene MF918561–MF918567, CNSV polyprotein 1 gene MK521453 and MK521454, 4349–4730 nt region of CNSV polyprotein 1 MK493788–MK493796, RdRp gene, subgroup B MK521443–MK521452.

Name of Cultivar	Abbreviation ^a^	UMNA Peony Collection Plant Accession Number	194 K RNA Polymerase Gene, 779 bp (TRV)	16 kDa Putative RNA Silencing Suppressor (TRV ^b^)	Polyprotein 1 Gene, Partial, ~450 bp (CNSV ^c^)	4349–4730 nt Region of CNSV Polyprotein 1, 380 bp (CNSV)	RdRp Gene, Subgroup B, 250 bp (CNSV)	RdRp Gene, Subgroup A, 340 bp (CNSV)	TRV Visual Symptoms (Severity, 0–5) ^d^
Duchess of Portland	DP1-2015	MBGNA-P-0147	−	−	na	+	+	−	0
Duchess of Portland	DP2-2015	MBGNA-P-0148	+	+	na	−	−	−	1
Gisele	Gis1-2015	MBGNA-P-0231	−	−	na	−	−	−	0
Gisele	Gis2-2015	MBGNA-P-0232	+	+	+	+	+	−	2
Gisele	Gis1-2016	MBGNA-P-0231	−	−	na	+	+	−	0
Gisele	Gis2-2016	MBGNA-P-0232	+	+	na	+	+	−	3
Yeso	Y1-2016	MBGNA-P-0710	+	+	na	+	+	−	4
Yeso	Y2-2016	MBGNA-P-0711	+	+	na	+	+	−	4
Gigantea	G1-2016	MBGNA-P-0227	+	+	na	+	+	−	3
Gigantea	G2-2016	MBGNA-P-0228	+	+	na	−	−	−	1
Jeannot	J1-2016	MBGNA-P-0289	−	−	na	+	+	−	0
Isoline	I1-2016	MBGNA-P-0272	−	−	na	+	+	−	0

^a^ Abbreviations used in Figure 2 and Figure 3. ^b^ Independent identification of TRV, using real-time PCR, of the 16 kDa putative RNA silencing suppressor by a commercial company (Agdia Inc.; see text). ^c^ Isolated through cloning. ^d^ Modified from [51] (see text for details).

## Data Availability

The original contributions presented in this study are included in the article and Appendix A; further inquiries can be directed to the corresponding author.

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
