# Peer review of "Co-Infection of Tobacco Rattle and Cycas Necrotic Stunt Viruses in Paeonia lactiflora: Detection Strategies, Potential Origins of Infection, and Implications for Paeonia Germplasm Conservation"

_viruses, 2024, doi:10.3390/v16060893_

Round 1

Reviewer 1 Report

Comments and Suggestions for Authors

Original article Co-infection of tobacco rattle and cycas necrotic stunt viruses in Paeonia lactiflora: detection strategies, potential origins of infection and implications for Paeonia germplasm conservation by Nastassia B. Vlasava et al. provides new valuable data on the severity and symptoms of tobacco rattle virus (TRV) and cycas necrotic stunt virus (CNSV) co-infection on plants of the peony collection. The materials are well structured and provide a qualitative analysis of other works.

The manuscript is formatted according to the rules and contains all the necessary sections, the bibliography is correct and provides a wide coverage of works in this area.

However, I believe that the manuscript needs not complex but fundamental changes.

The data given in the supplement should be transferred to the text of the main manuscript, possibly with the exception of gels and their descriptions. The gels must contain the correct size of the fragments, the markers used, and it is better that they be next to the images of the symptoms.

I also recommend adding a description of the symptoms depending on the leaf shape, veins, and manifestations on the leaves in different parts of the plant. It is advisable to put bars in the photographs. Since, for example, the ratio of spots is not clear, taking into account the fact that you present varieties that differ greatly in the shape and size of the leaf blade.

Also, the methodology should reflect at what stage of plant development the identification of the viral infection was done and recommended (from the authors’ point of view) (and why).

The description of the method does not describe extraction, however, this is a methodologically key point for the analysis of nucleic acids, especially those with complex localization. Please clarify this.

I recommend that you provide the sequences of all the primers that you used in your work.

Given the original comparative approach, can the authors suggest ways in which viral infections may spread due to this focus on the geolocation of comparisons?

The article may be published after modification.

Reviewer 2 Report

Comments and Suggestions for Authors

The authors have identified the plant materials, Paeonia, to investigate the presence and further characterization of both tobacco rattle virus (TRV) and cycas necrotic stunt virus (CNSV). I must say that this is an interesting observation. The main findings were the confirmed infection of both viruses, including four plants of co-infection, showing more severe disease symptoms than plants only infected with TRV. The authors further explored the origins of these viral infections based on phylogenetic analyses. The authors claimed that this is the first report of TRV/CNSV co-infection and symptomatic detection of CNSV in P. lactiflora cultivars.

I believe that the authors provided sufficient background, explained the methodologies well, presented the data using appropriate tables and figures, and concluded appropriately based on available data. The overall presentation is fine with acceptable language (English), and I only see a few grammatical and editorial errors throughout the entire manuscript. I only have a few minor suggestions (listed below) for the authors to consider if a revision is requested by the editor.

Introduction:

I believe that the authors did a good job explaining both viruses and explicitly described the goals of this study, though I would suggest that the authors use the word “goals” or “objectives” somewhere in the last paragraph of Introduction.

Materials and Methods:

I believe that the authors have explained very well the methodologies used in this study for reproducibility. To be consistent, I would suggest that the authors use the past tense in this section.

Table 1: the bottom of this table need a visible horizontal line.

Table 2: I would suggest that the authors move Table 2 to the place after Line 176, because it is where this table is cited.

Line 198: please delete “Fraser and Loughlin (1980)”

Figure 1: is it possible that the authors update the figures of viral infection just because these pictures shown here were taken almost 10 years ago? This is important to know the current status of these viruses identified about 10 years ago. Please make sure to use present tense in figure legend; this is the same problem for other figures.

Results:

I appreciate very much the well-organized figures, which help a lot to understand the results of this study.

Table 2: the display of this table is strange, needs to be re-made, in particular the top two rolls, which may be simplified to improve the visibility of this table.

Discussion:

The authors provided in-depth discussion of results and concluded appropriately based on available data.

Line 355: Do authors have any evidence to support the statement “the transmission (possibly by nematodes)”?

Line 380: genes but not gens?

Line 398: I would suggest that the authors further elaborate on the significance of the first report on the identification of the co-infection of these two viruses.

Comments on the Quality of English Language

English is acceptable, but needs a minor revision, as I indicated in my review for the authors.

Reviewer 3 Report

Comments and Suggestions for Authors

The manuscript “Co-infection of tobacco rattle and cycas necrotic stunt viruses in Paeonia lactiflora: detection strategies, potential origins of in fection and implications for Paeonia germplasm conservation” submitted by Vlasava et al., was carefully reviewed. Tobacco rattle virus (TRV) and cycas necrotic stunt virus (CNSV) of Paeonia lactiflora are common plant diseases, which seriously affects the growth and development of P. lactiflora. The authors report the molecular characterization of TRV and CNSV in living peony collection and first reported the TRV/CNSV co-infection and symptomatic detection of CNSV on cultivated P. lactiflora. Data collection is sufficient, which has important reference value for disease resistance breeding and production of P. lactiflora.

My main concerns are as follows:

Each small figure in Figure 1 and Figure 4 should be supplemented with an independent bar.

The English unit “μl” needs to be replaced by “μL”

For table 2, the name of the sample is too complex, because it is not the number recognized in the market or researchers, it should be the mark of the author's experimental process. In this table, it should use a simpler abbreviation and note the relevant meaning clearly.

Round 2

Reviewer 1 Report

Comments and Suggestions for Authors

I believe the authors have corrected the comments made. The manuscript can be accepted into the Viruses journal